# Sex trafficking survivors' experiences with the healthcare system during exploitation: A qualitative study

**Johane Lorvinsky**[1☯], **John Pringle**[1,2☯], **Françoise Filion**[1☯], **Anita J. Gagnon**[1,3☯]*

**1** Faculty of Medicine and Health Sciences, Ingram School of Nursing, McGill University, Montréal, Québec, Canada, **2** Médecins Sans Frontières, Geneva, Switzerland, **3** Research Institute, McGill University Health Center, Montréal, Québec, Canada

☯ These authors contributed equally to this work.
* anita.gagnon@mcgill.ca

**Data Availability Statement:** We have consulted our ethics review resources and based on the language on the consent forms, we cannot share data through repositories or otherwise. For more information interested researchers may contact

## Abstract

While most individuals who have experienced sex trafficking will seek medical attention during their exploitation, very few will be identified by healthcare professionals (HCP). It constitutes a lost opportunity to provide appropriate support, resources, and services. In this study, we examined the experiences of accessing care of sex trafficking survivors in the Greater Montreal area and their interactions with HCPs to inform trafficking education programs for HCPs and allied health professionals regarding the needs of this patient population. We conducted seven semi-structured in-depth interviews with purposively selected sex trafficking survivors participating in "Les Survivantes," a program of the SPVM (Service de Police de la Ville de Montréal), designed to support trafficked individuals' exit journey. We used interpretive description to understand the lived experiences of trafficked individuals with direct applications to clinical education and care. Our results revealed that trafficked individuals accessing care present with a fragile trust in HCPs and how HCPs have many opportunities to conduct comprehensive examinations and query trafficking. Trafficked individuals' initial trust in HCPs can be strengthened by non-judgemental approaches or damaged by stigmatizing conduct, serving to isolate further and alienate this patient population. Health professionals' attitudes combined with healthcare settings' cultures of care (i.e., community vs emergency) and exposure to marginalized groups were key influencers of survivors' perception of healthcare interactions. The findings also emphasized the importance of routinely querying trafficking through sensitive psychosocial questioning based on observation of trafficking cues. Survivors reported a list of trafficking cues to recognize and emphasized the importance of trust as a condition of disclosure. Finally, survivors identified the need for exit planning to be centered around trafficked individuals' agency and holistic needs, and for streamlined community-based multidisciplinary collaboration to better serve this population. Our results highlight that most challenges experienced by trafficking survivors in accessing care and resources are modifiable through HCP education and training. Our study also provides new insights and concrete advice to improve care and support throughout the exiting process. We argue that healthcare services for this population be modeled harm reduction approaches that focus on victims' agency and needs, independent

sacha.young@mcgill.ca (McGill University, Faculty of Medicine Institutional Review Board).

**Funding:** The author(s) received no specific funding for this work.

**Competing interests:** The authors have declared that no competing interests exist.

of their desire to exit trafficking. We emphasize the urgent need for proper case management and intersectoral and multidisciplinary care coordination in community-based settings as well as facilitated access to mental health support.

## Introduction

According to the International Labour Organization's most recent and rather conservative estimates, 89 million people were victims of human trafficking worldwide between 2012 and 2017, with 40.3 million actively being trafficked in 2016 [1]. In Canada, the consensus is that human trafficking in and through the country is much more prevalent than suggested by previous estimates or the 1700 police-reported cases from 2009–2018, of which commercial sexual exploitation or sex trafficking is the largest subset [2]. Underreporting can be attributed to the hidden nature of the industry [2], unawareness of victimization [2] or fear and distrust of authorities [2,3], and variations in sex trafficking definitions (e.g., conflation with sex work) [4]. This study adopts Macy & Graham's (2012) definition of sex trafficking as "The recruitment, harboring, transportation, provision, obtaining, patronizing, or soliciting of a person for the purpose of a commercial sex act, in which a commercial sex act is induced by force, fraud, or coercion, or in which the person induced to perform such an act has not attained 18 years of age" [5].

### Review of the literature

Individuals in sex trafficking experience a wide array of physical, sexual, and psychological violence [6–8]. They commonly report incidences of assault, threats, intimidation, humiliation, and degradation, as well as rape and restriction of freedom [9]. Sequelae from these harms are wide-ranging and include several physical, mental, and reproductive health problems [6,9–11]. Most commonly, individuals under sexual exploitation experience acute injuries, untreated health conditions, anxiety and depression, post-traumatic stress symptoms, sexually transmitted and blood-borne infections (STBBIS), numerous pregnancies and terminations (spontaneous or induced), and other sexual health issues [6,9–11]. The significant medical and mental health needs may result in their seeking medical treatment [9,12]. Research in other developed countries has shown that trafficked individuals have limited access to healthcare as they are typically monitored, isolated, and denied access to their identifying documents such as their health insurance cards [5,6]. Yet, most trafficked individuals receive medical treatment at some point during their exploitation [9]. These points of contact with healthcare are often missed opportunities for healthcare providers (HCP) to identify trafficked individuals and provide services and resources to address their needs [9,13]. When they meet HCPs, trafficked individuals seldom disclose their abuse due to fear of their trafficker, low self-esteem and feelings of shame, and lack of knowledge about judicial and healthcare systems [5,6]. Concurrently, HCPs are ill-prepared to identify trafficked individuals [9]. Under-identification means less access to services and resources to exit the trafficking situation [5]. The lack of evidence-based guidelines or protocols on how to engage with and care for this population, as well as the limited resources to meet the needs of survivors, further alienate trafficked individuals and may result in impaired future health outcomes for survivors [10,14]. Training for HCP needs to include clear guidance to identify people in trafficking and steps to intervene with information on relevant resources and specialized trafficking services [15].

The program "Les Survivantes" was created by officers from the Section des Enquêtes Multidisciplinaires et Coordination Jeunesse (SEMCJ) within the Service de police de la Ville de

Montréal (SPVM) to raise awareness of trafficking and to support and educate service providers that interact with sex-trafficked individuals. They provide information to other police officers, workers from youth centers, and HCPs, and rely on the expertise of survivors. The program also provides resources and information for youth-at-risk and their families and accompanies survivors of commercial sexual exploitation in their exiting journey by facilitating access to community services and assisting them with legal processes if they choose to press charges. As awareness of sex trafficking develops, so does the need for quality, evidence-based research to inform identification, intervention, and prevention. In conjunction with the directors of "Les Survivantes" program, we conducted this needs-based study in Montreal, Quebec to explore the experiences of sex trafficking survivors with the healthcare system. Hearing from survivors exiting sex trafficking and positioning them justifiably as experts provides insight into their lived experiences to inform future education programs for HCPs tailored to the needs of this underserved population. Of particular interest are survivors' ability to perceive, seek and engage in healthcare, the missed cues such as signs and symptoms that may have been conveyed to healthcare workers indicating that their patient is being sex trafficked and their advice for HCP intervention once identified. Missed cues represent missed opportunities to meet the person where they are in their journey, and a barrier to provide appropriate care and to promote liberation.

Few studies have sought to explore trafficking survivors' experiences with the healthcare system in Canada during their exploitation [16], and to the authors' knowledge, none were specific to the Greater Montreal area. Since experiences of sex trafficking are often specific to population demographics and the larger societal context/geographic region, this study will equip providers to recognize trafficking cues specific to the Greater Montreal, an area cited as a destination for sexual tourism in North America [17].

## Methods

### Study design

We conducted qualitative research using an interpretive descriptive design [18] to explore the experiences of sex-trafficked survivors with the healthcare system during their exploitation. Interpretive description is an inductive analytic approach designed to create ways of understanding clinical phenomena that yield direct applications to clinical care [18,19].

### Study population and sampling

The study population was sex-trafficked individuals in Quebec, with the sampling frame comprised of a list of Montreal-based survivors who have used the services of the Survivantes program to completely exit trafficking. The study eligibility criteria included having exited sex trafficking within the last 15 years; having had an interaction with the healthcare system during exploitation; currently 18 years of age or over; able to provide informed consent; and being able to speak English or French. We used purposive sampling. Eligible Les Survivantes members were notified of our study, and only those who expressed an initial interest in participating were contacted by the Principal Investigator (PI) for recruitment.

### Data collection

We conducted single individual semi-structured in-depth interviews by video or teleconference, between September and December 2020. Our interview guide (see Appendix A in S1 Appendix) was validated by our research partners at SPVM and a survivor in the Survivantes Program who did not participate in the study. Participants were asked about the context of

their interactions with the healthcare system during their exploitation and their perception of care received. We also sought from them practical advice for HCPs working with sex-trafficked individuals. Corresponding socio-demographic information was collected by way of a short questionnaire (see Appendix B in S1 Appendix) at the start of the interview. The interviews were audio-recorded and transcribed verbatim by the PI and a professional transcriptionist. All transcripts were also assessed for accuracy. Quotes and citations were machine-translated and edited by PI who is fluent in both languages (see Appendix C in S1 Appendix).

## Data analysis and rigour

Data collection and analysis were conducted concurrently, in conformity with interpretive description. Transcripts were analyzed using NVivo software QSR International Pty Ltd. (2020), with open or line-by-line coding, inductive reasoning, and comparative and iterative analysis. Transcripts were broken into segments (codes) and compared to other segments to identify commonalities and variations [19,20]. Patterns were identified and data were separated into categories, then themes [18,19]. To address the four constructs of trustworthiness [21], the PI maintained complete records of all collected materials and documentation and used methodological triangulation (i.e., observations and interview transcripts), transcription rigor, and regular peer debriefing throughout data collection, analysis, and reporting. The PI also researched the context of the phenomenon for a year before conducting the interview.

## Ethical considerations

The research protocol received research ethics approval from the McGill University Faculty of Medicine and Health Sciences Institutional Review Board (Study No. A06-B36-20B). The principal investigator obtained informed consent from each participant before commencing the interview and administering the demographic questionnaire. Participants were informed of their right to stop the interview, to refuse to answer questions and to retract their participation at any time throughout the interview without any repercussions. To protect confidentiality, ID numbers were used in place of participants' names on all recordings, transcripts, and other documentation. No names appear in this publication, and no identifying information is divulged. Care was taken to prevent and mitigate distress during the interview (see Appendix D in S1 Appendix) and participants had access to mental health care through the survivors' program. They were strongly encouraged to utilize them if they needed additional support post interview.

## Results

Of those potentially eligible in Les Survivantes program as determined by our SPVM research partners, twelve women expressed an initial interest in participating in the study. However, of the twelve women, two did not return our call, two accepted but did not present to the interview, and one was deemed ineligible. Therefore, a total of seven women participated in this study, a sample size that was sufficient to achieve saturation (informational redundancy). Participants' socio-demographic data are reported in Table 1. Participants were between the ages of fourteen and twenty-four during their trafficking period and had previous history of dysfunctional family processes. Six out of seven reported being in foster care or unhoused at the time of grooming, sex work, or beginning of trafficking, and two reported physical abuse in their childhood. No participants had contact with family members during their trafficking period.

Participants' accounts revealed the following themes: (1) Advantages of healthcare settings; (2) HCPs' attitudes affecting care; (3) Trafficking recognition and disclosure; and (4) Exit

**Table 1. Characteristics of study sample.**

| Characteristics | Participant (n = 7) | |
| --- | --- | --- |
| | *N* | *%* |
| Age (range) | 24–36 | |
| Median (SD) | 32 (4.2) | |
| Ethnicity | | |
| Caucasian | 4 | 71 |
| Indigenous | 3 | 43 |
| Level of Education | | |
| High school | 4 | 57 |
| CEGEP or professional school | 3 | 43 |
| Visible minority | | |
| Yes | 2 | 29 |
| No | 5 | 71 |
| Disability | | |
| Yes | 1 | 14 |
| No | 6 | 86 |
| **Trafficking related data (n = 7)** | | |
| Age of onset | | |
| 14 | 1 | 14 |
| 16 | 2 | 29 |
| 17 | 1 | 14 |
| 18 | 2 | 29 |
| 21 | 1 | 14 |
| Duration | | |
| <5 | 4 | 57 |
| > = 5 | 3 | 43 |
| Exit LOT | | |
| <5 | 1 | 14 |
| > = 5; <10 | 1 | 14 |
| > = 10; < = 15 | 5 | 71 |

[a]Visible minority: Non-white in skin color.

[b]CEGEP = Collège d'enseignement général et professionnel (similar to a technical college or vocational school).

[c]LOT = length of time.

support. The themes are not mutually exclusive. Themes (blue) and sub-themes (white) are reported in Fig 1.

## Theme 1: Advantages of healthcare settings

This theme captures participants' accounts of why encounters with the healthcare system during exploitation are critical and unique opportunities for intervention and support for sex-trafficked individuals. Despite limitations in access to care, most participants who went through the healthcare system had a positive initial perception of healthcare and HCPs, were generally not accompanied by traffickers, and underwent physical assessments that could reveal abuse and lead to further inquiry.

**Variation in access to care.** Survivors interviewed reported varying degrees of freedom of care, which was largely determined by the limitations imposed by traffickers. Most related how

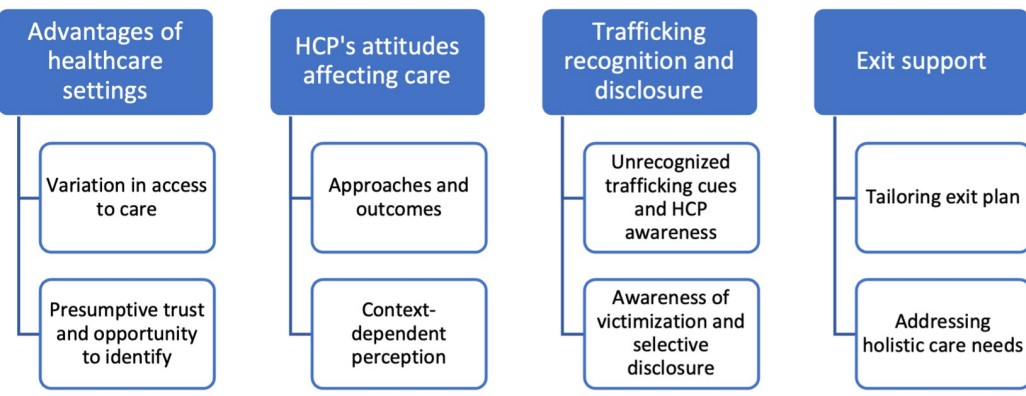

**Fig 1. Experiences of trafficked survivors with the healthcare system.**

traffickers exerted power and control over their decisions, including their ability to seek care, through a variety of nefarious means such as violence, threats, continuing monitoring, financial control, and gaslighting. These tactics led to a loss of autonomy for participants, who were unable to make their own decisions, including seeking any form of help beyond the limitations imposed.

> If you are under the influence of someone, you can't make your own decisions, your life is governed by another person, and if you are being abused, the person will always ask you where you are, what you are doing, and who you are with.

> (P2)

In one case, the survivor was completely banned to seek medical treatment and their interaction with healthcare was accidental: "Well, I went straight to the hospital because I was in Quebec City. . ... and it was a car accident because otherwise, I would never have been able to go to the hospital" (P1). Others were allowed to seek treatment for emergencies such as when a health issue impeded trafficking activities or in times of severe crisis, or to maintain their sexual health. As participant 5 stated, "If you're run down, he has no choice but to take you to the doctor. Otherwise, you don't make money,". Some survivors were able to have regular access to a community-based organization offering safe drug paraphernalia and sexual health services for STBBI prevention and treatment: "they expect you to have good sexual health because they won't stop you from getting checked" (P2). These contact with healthcare were still under close monitoring for some survivors, through constant text messaging or fear of repercussions if they took too long to return. For one survivor, the trafficker went as far as accompanying her to the hospital. Survivors who did not relay explicit limitations to healthcare utilization also mainly sought treatment for sexual and reproductive health emergencies. Details of the context of healthcare experiences can be found in Table 2. Participants agreed that most sex-trafficked individuals will seek healthcare during their exploitation given the risks inherent to sex work.

**Presumptive trust and opportunity to identify.** Regardless of the conditioning of traffickers to mistrust law enforcement and even their peers, survivors hinted at a level of presumptive trust in healthcare systems and HCPs. Participant 1 explained that trafficked individuals are "brainwashed" by traffickers to fear law enforcement and unable to trust other sex workers/trafficked individuals, and therefore, "the trusting relationship is important because often these girls don't have any" (P1). Many stated that hospitals represented safety,

**Table 2. Context of healthcare experience.**

| Participant | Setting | Presenting issue/Reason for seeking care |
|---|---|---|
| 1 | Emergency care | Motor vehicle accident |
| 2 | Local Community Service Centre | STBBI treatment |
| | Emergency care | Drug overuse |
| 3 | Emergency care | Sought a Sexual assault forensic exam |
| | Emergency care | Drug addiction |
| | Addiction rehabilitation center(1) | Drug addiction |
| 4 | Emergency care | Miscarriage |
| 5 | Walk-in clinic | Anxiety attack (e.g., thought she had AIDS) |
| 6 | Local Community Service Centre | Sexual assault forensic exam |
| | Local Community Service Centre | Health insurance card replacement |
| 7 | Local Community Service Centre | Abortion |
| | Community-based organization | Contraceptives and drug tools |
| | Unspecified doctor's office | Intrauterine device insertion |
| | Unspecified doctor's office | STBBI screening |
| | Youth protection | Contact with social workers |
| | Midwife private practice | Pregnancy/abortion |
| | Hospital | Abortion |

security, and assistance, and doctors and nurses were seen as credible figures of authority. Consequently, these participants expressed being more likely to disclose sex trafficking to HCPs than to police officers. Participants also added that healthcare settings come with a wealth of opportunities for HCPs to be alone with trafficked individuals and to examine their bodies, which makes them more likely to identify signs of abuse and trafficking and pursue further questioning. Furthermore, most survivors accessed care unaccompanied. "The [trafficker] is not going to wait six hours in the hospital with you, so you're going to be there by yourself. That makes it the right time to try to get the person to do something" (P4). And even when trafficked individuals are accompanied by the trafficker, like Participant 3 was, HCPs can require examining their patient privately. "I don't think that anybody else should be in the room when you go see your doctor" (P3). So, despite traffickers' power over survivors being their main barrier to accessing care, HCPs have a distinct power and opportunity to assess, identify, and help them exit trafficking.

## Theme 2: HCP attitudes affecting care

This theme examines survivors' perception of their reception by and attitude of the HCPs, the outcomes of the interactions, as well as advice on how to approach individuals who may be experiencing trafficking.

**Approaches and outcomes.** Findings indicated that perceived behavior and approach of HCPs were determining factors in health outcomes for this patient population. Women who had positive HCP encounters reported feeling "supported", "reassured" and "helped". Being provided compassionate and non-judgmental care led them to develop trusting relationships that strengthened health-seeking behaviors and encouraged disclosure of sex work. Such an approach led one participant to disclose sex work to nurses at her local community health center:

Because I trusted them because I was already going there on a regular basis to get STD tests or to get like condoms (. . .) I was honest, I told them as soon as I started [speaking of sex work]. There was no judgment on their part".

(P2)

On the other hand, unpleasant interactions were characterized by cursory care and by judgmental, dismissive, cold, or indifferent HCPs. Participants' accounts revealed that their perception of encounters with HCPs was strongly impacted by experiences of stigma. Women with unpleasant experiences when seeking care commonly reported HCPs making disapproving comments, criticizing, gossiping, or judging their behaviors. Some felt like they were treated like burdens, unwelcomed, and felt like HCPs were quick to label, victim-blame, and pathologize them. Common subjects of stigma were sexual behaviors, abortion, STBBI, and drug use. Stigmatizing attitudes, prejudices, and actions by HCPs only served to damage initial trust, reinforce trafficked individuals' negative feelings about themselves, and create or confirm fear and anticipation of judgment. Participants pointed out that women in trafficking might already be carrying a lot of shame and are generally scared of judgment, so further shaming when accessing care is additionally detrimental. "You already self-flagellate yourself, so the judgment of others on top of it, forget about it." (P7). For some these experiences reinforced their belief that no one can help them: "I left those situations, feeling helpless, scared and there was nothing I could have done (. . .) I don't want people to leave feeling even more helpless than when they came in" (P3). A further surprising finding is that one participant related experiencing breached sexual boundaries with care providers in the form of inappropriate sexual humour or demeaning comments during visits, and even developing sexual relationships with caseworkers. According to her, because sex workers are used to pandering to the male gaze, they are more attuned to HCP sexual interest, but a violation of sexual boundaries inevitably leads to damage of trust. "It kind of defeats the picture of an ally when he wants to sleep with you and then he says he wants to help you, f-you. Like. . . you're like another one of them." (P7). These negatively perceived interactions led to fragmented or inadequate care and outcomes such as premature discharge, non-disclosure, alienation from the healthcare system, or feelings of helplessness and shame. Participant 5 also added that it is harder to trust HCPs after a negative experience.

**Context-dependent perception.** Participants' perception of their healthcare experiences was also influenced by the setting in which they accessed care. In more acute care settings such as hospitals or walk-in private clinics, participants were more likely to have experienced impersonal, hasty, and perfunctory care. In these settings with lower exposure to marginalized groups most associated with being trafficked (e.g., youth in precarious situations, such as unhoused or with drug dependencies), HCPs were also seen as more prejudiced. Conversely, higher exposure settings such as community care centers, sexual health clinics, or community-based organizations offering sexual health services (e.g., STBBI testing, contraceptives, etc.) had mostly positive reviews. Participants theorized that HCPs in these settings were more progressive and trained to provide compassionate and non-judgmental care for marginalized groups. Therefore, participants felt more at ease accessing care through these community-based care services to avoid stigmatizing situations. "The fact that they are in contact with this type of clientele [sex workers], was more conducive to. . . you didn't need to talk about it, they knew it" (P7). Perception of HCP in different healthcare settings can be found in more detail in Table 3.

These findings indicate that a non-judgmental approach—one that builds rapport and a trusting relationship—is essential for trafficked individuals to maintain a positive perception

**Table 3. Perception of Healthcare providers (HCP) by healthcare setting.**

| Healthcare setting | Perception of HCPs | Quotes |
|---|---|---|
| Emergency Room | Nice, reassuring | "Well, they were normal. Super nice though, as I recall" "I could see that the professionals were very gentle, so it must have seemed that I was nervous" (P1). |
| | Judgmental and critical | "I had told them [HCPs] how I had been using and then they would hint that it was bad and then... but I didn't need judgment at that point, I just needed to get help" (P2). |
| | Dismissive and victim − blaming | "They were treating me like I was some sort of f-ed up burden of some sort" (P3). "she got raped and she deserves it, she put herself there so why are we going to help her? And it's victim-blaming a lot" (P3). "the door was open and when I asked them to close it completely, they wouldn't close it completely. They left it a bit open" (P3). |
| | Distant and indifferent | "The way they were, they were really indifferent and then no one asked me any questions... you had a miscarriage we're just going to go check if there's anything left inside and then go back home" (P4). |
| Maternal care and pediatric hospital | Judgmental | "I remember that [Hospital Name] was not great, because you know there is a lot of judgment on abortion and all that and even you judge yourself and you don't need that from others"(P7). |
| Walk-in Clinics | Rushed, Distant | "Zero questions. They gave me a prescription for pills, it lasted three minutes. He gave me a prescription for pills, a paper to do a blood test and I remember I was crying, then I said 'Well I've never done a blood test, where am I going? I don't even have a health insurance card '" (P5). |
| Local Community Health Center | Open and understanding | "It's something that's felt, but it was like I had a flu, it was no more serious than that. Just understanding with me and there was always a gentle tone and then there was no judgment" (P2). "There was a CLSC that was more open, (....) that's when I had my abortion, they were used to working with people who are prostitutes and all that, so they were more open." (P7). |
| | Respectful | "They [HCPs] helped me. They treated me with respect, I didn't feel racism when I went there... I didn't feel targeted, "ah she's selling her ass, she's using", even in a drugged state they were helping me" (P6). |
| | Judgmental | "The nurses were sometimes judgmental and that's being human I think." (P7). "I remember seeing the nurse afterwards and she said do you regret taking the baby out" (P7). |
| Community-based Organization | Trustworthy, Non − judgmental, Sex − worker friendly | "There it was very inclusive; I loved their approach. That's where I also went to get condoms.... there were a few organizations downtown that I went to regularly...The fact that they are in contact with this type of clientele [sex workers], you didn't need to talk about it, they knew it" (P7). |
| Drug rehabilitation center | Attentive | "I kind of felt that when I was in the room, like, before like he concluded that rehab would be a great option for me, like he actually listened to me and wasn't rushing me. He wasn't judging me, and he was just there like tell me everything you want, or you have to, and what's going on with your health and what's happening because obviously something is going on and it's something that I can't go out and discharge you" (P3). |
| Midwife Clinic | Supportive, open | "She [Midwife] was really supportive and all that, she told me the day you want to get pregnant again [Come back]"(P7). |
| Youth protection center | Non − ethical, judgmental, indifferent | "At times it seems like you don't necessarily feel judged, not evaluated, it seems like they need to write a report. So, it's that impersonal side that's a little bit" (P7). "I've had relationships with case workers...It kind of defeats the picture of an ally when he wants to sleep with you and then he says he wants to help you, f-you. Like... you're like another one of them"(P7). |
| Unspecified doctor's office | Rude, non − ethical, rushed, judgmental | "He [HCP] asked me questions about, you know, yes I was having a lot of sex at that time, and then he said ah you're a horny bunny" (P7). "The doctors for me were five minutes and then you move on to the next" (P7). |

of healthcare, to seek help, and to feel safe enough to disclose abuse, sex work, or trafficking. For participants, such an approach requires HCPs to show respect, kindness, attentive listening, genuine interest in their patients, open-mindedness, and continuity of care. Participants also advise HCPs to refer trafficked individuals to someone else who can support them properly if they feel unable to provide non-judgmental care because of prejudicial beliefs or if they are unable to maintain ethical relationships with this patient population.

### Theme 3: Trafficking recognition and disclosure

This theme encompasses factors that impacted trafficking recognition and survivors' advice on how to assess and identify trafficking. Despite HCPs' opportunity to assess for abuse and trafficking, no participants were identified through their healthcare interactions. Findings pointed to multiple factors that prevented identification of sexual trafficking: HCPs' myopic focus on physical health and lack of awareness and education on trafficking from both HCPs and trafficked individuals.

**Unrecognized trafficking cues and HCP awareness.** Most women reported that HCPs asked an inadequate number of questions to reveal trafficking and were short-sighted in their assessment, going right to the physical problem at hand and neglecting psychosocial questions, or questions of abuse, home life, sex work, or trafficking. What is more, most participants felt trafficking cues were obvious during their interactions and that missed cues were mainly due to unawareness of trafficking. Survivors identified several signs of trafficking that were present during their healthcare encounters, including visual, behavioral, and contextual cues. Visual cues included marks of physical abuse (e.g., cigarette burns, bruises) and/or sexual abuse, and dress code (e.g., "stripper clothes"); behavioral clues were more subtle and included being anxious, stressed, fearful, pressed, and constantly texting; and lastly, contextual cues were varied and included atypical reasons for seeking care (e.g., anxiety regarding HIV/AIDS or request for rape kit test), stories inconsistent with injuries and dynamics with other persons accompanying trafficked individuals (e.g., constantly looking at someone, seeking approbation, someone answering or refusing tests and procedures for a patient). Individually, these signs may be insufficient to indicate trafficking, but in combination, they ought to raise alarms. Yet, unawareness precludes recognition: "[HCPs] who are not aware [of sexual trafficking] and who do not know that it can happen, (. . .) it is certain that you cannot guess things when you are not aware that they exist" (P1). Once aware it's easier to pick up on clues. Awareness was also identified by survivors as a key to lessening stigma and misconceptions of the realities of trafficking. Some misconceptions are detrimental to trafficking recognition, namely the assumption that involvement in sex work is always voluntary. For instance, one survivor who was a known sex worker to community nurses was never asked about working conditions, safety in her workplace, or desire to exit because "when you say sex worker, in their minds it's voluntary" (P2). Although most participants felt that awareness about trafficking had increased in recent years, accounts were similar whether they exited 2 years ago or 15 years ago, especially regarding stigma, misconceptions, and missed identification in their interaction with the healthcare system. Some participants mentioned having interactions with HCPs in which they believed the HCP had an inkling that something was wrong but did not act on their suspicions. One woman remarked that as an HCP, it can be uncomfortable and difficult to ask trafficking- or abuse-related questions and suggested the need to support HCPs who are apt to hear about traumatic events. Other participants also stressed the need for HCP to be trained on the realities of people in trafficking and equipped to recognize cues. They also offered advice on how to frame the questions. Notably, they cautioned against direct questions like "Do you have a pimp?" (P1). Instead, they suggested sensitive and respectful psychosocial questions relevant to

the case and based on observations like asking about work conditions, or presence of violence or exploitation in their work and relationships. Upon recognition of trafficking cue, asking is crucial: "Ask them the question. The only thing that can happen is that you will insult them. At least you won't have missed something (. . .) you might save someone" (P5).

**Awareness of victimization and selective disclosure.** Survivors' accounts of healthcare interactions reveal that disclosure is dependent on many factors, notably the individual's self-awareness of trafficking. Participants discussed being unaware they were being exploited at first. For most, it took them years to realize that their situation and relationship(s) were not normal. One survivor explained that most believe it cannot happen to them, although it can happen to anyone regardless of background and status. She also remarked that transition from sex work to trafficking can be subtle:

> I was lying to myself as well because I think that in the beginning that's what the people you're working for make you believe, but it's sure that with time (. . ..) you give them half of your money, you give more than half of your money and then it's sure that at a certain moment they make you live things-, well with time, I think it became exploitation but even me, in the beginning, I thought it was sex work.
>
> (P2)

Trafficked individuals' unawareness is a barrier to disclosing trafficking and indicates the need to raise awareness within the sex worker population too. Participants agreed that HCP's query of sex trafficking could lead sex-trafficked individuals to realization and suggested other ideas for awareness initiatives such as posters in hospital waiting rooms. Other factors that impact likelihood of disclosure are trusting relationships and pertinence of disclosing. Many expressed that, subjects such as drug use, sex work, abuse, and trafficking are hard to disclose as disclosure might invite moral judgments from HCPs and lead to shame and embarrassment. However, most participants stated that they might have responded if asked by someone with whom they were comfortable and found it more pertinent to disclose sex work when it increased access to goods or services offered to sex workers (e.g., contraceptives or drug instruments). Considering these conditions for disclosure, and with awareness and training, HCPs can recognize the need to query sex trafficking and facilitate sex-trafficked individuals' self-awareness.

## Theme 4: Exit support

According to participants, exiting trafficking is a difficult process that can take years, and some survivors can relapse many times before doing so completely, especially when proper exit support is not provided. Therefore, this theme encapsulates advice from participants on how to provide high-quality exit support to facilitate the process. They highlighted the necessity to assess needs and resources already available to sex-trafficked individuals to tailor exit planning and to offer services to address holistic care needs.

**Tailoring exit plan.** Once identified, participants remarked on the importance of tailoring the approach and interventions to survivors' unique characteristics and needs. As one participant stated: "I think having a good support program, you tailor it to each individual, try to figure out what's their home situation with their family like, you know, is there any resources that them or their family could have" (P3). There is no one-size-fits-all solution, "it's a matter of working with that individual, that person" (P3). Survivors highlighted the need for HCPs to evaluate and clearly define patients' holistic needs to plan the most appropriate management course. That may include mental health support, detoxification, or being followed by a

resource person to arrange care services or social support, housing, and finances. One participant cautioned HCPs against prescribing medication too hastily without managing underlying issues first, as it led her to develop a dependence that might have been prevented by proper mental health support: Participants also mentioned the imperative to assess readiness for change and exit, and to avoid imposing help on those who are not ready for it. Sex-trafficked individuals might not be ready to leave their lifestyle because of fear of uncertainty or lack of social support. For that reason, exiting may be a long process fraught with relapse into drug addiction, sex work, or revictimization.

**Addressing holistic care needs.** Nevertheless, the HCP's role is to offer support and safety along with resources and healthcare options that survivors can access when ready. Several participants noted having no knowledge of exiting services for sex-trafficked individuals or difficulty in navigating the healthcare system to access existing resources. They suggested that HCPs should give resources and referrals to services, regardless of desire to exit. For instance, Participants 4 and 7 suggested giving pamphlets, or other discreet communication mediums to avoid trafficker's suspicions, pointing to existing services to sex-trafficked individuals who are not ready to disclose or exit trafficking. Those ready to exit trafficking, should be offered a concrete plan of intervention responding comprehensively to their needs. They must have a strong sense of safety to leave everything they know, As Participant 3 stated:

> "For me to trust someone like, I would rather run back to my pimp than have to go through so many different loopholes just to get where I need to get to be safe. I would rather just go back there because I know I have a place to stay, I know I'll have food, and I know I'll have like certain things. Yeah, I might have to be abused but I don't want to deal with uncertainty. I want to go into a situation feeling safe and confident.
>
> (P3)"

Most commented on the lack of mental health support available to them during exploitation and reported the need for mental health care providers trained to help individuals who have experienced trafficking and abuse. Participants specified that these services need to be accessible without referral, free, and should include drug addiction counseling. Participant 2 explained that having access to mental health services might mitigate some of the damaging effects of trafficking on survivors. Participants also mentioned the need for continued outreach programs to promote condom use, STBBI prevention, and safe drug use. Outreach initiatives could also benefit from the experiential knowledge of survivors to help identify and better relate to this population. Survivors commonly felt that having gone through trafficking, they were able to recognize other sex-trafficked individuals and connect on a personal level with them. More survivor's advice can be found in Appendix E in S1 Appendix.

## Discussion

This study explored sex trafficking survivors' interactions with healthcare during their exploitation. Four key themes emerged from the data: Advantages of healthcare settings; HCPs' attitudes affecting care; trafficking recognition and disclosure; and exit support. Our results highlight the challenges experienced by sex-trafficked individuals in accessing care and provides new insights on how to improve care and support them throughout the exiting process. Recommendations were derived from survivors' advice and are summarized in Table 4.

The emerging themes are consistent with existing reports of survivors' interactions with the healthcare system, such as missed opportunities to assist sex-trafficked individuals' exit because of numerous barriers that include a lack of trafficking awareness and training on the

**Table 4. Summary of recommendations for healthcare providers (HCPs) and organization.**

| | HCP Level | Organization Level |
|---|---|---|
| **Education** | • Identify personal biases and misconceptions about sex trafficking, sex work, and drug use.<br>• Seek out educational opportunities regarding sex trafficking.<br>• Share knowledge about sex trafficking with colleagues | • Include training on sex trafficking in HCP's curriculum and in the workplace.<br>• Training should cover:<br> • Cues to trafficking and differentiation with sex work.<br> • Approaches with sex-trafficked individuals (e.g., harm reduction, trauma-informed care, and cultural competence).<br> • Screening for sex trafficking.<br> • Exit planning.<br>• Utilize community stakeholders and survivors as speakers. |
| **Screening** | • Integrate harm reduction, trauma-informed care, and cultural competency into standard practice.<br>• Routinely query psychosocial wellbeing and assess for signs of abuse and trafficking.<br>• If signs are present, frame questioning on observations and ask about trafficking.<br>• Facilitate victim awareness and disclosure. | • Develop screening tools for identification. |
| **Interventions** | • Offer assistance and services; implement protocols and care pathways, independent of desire to exit sex work.<br>• If not ready to leave trafficking, provide them with information about sex work community organizations, harm reduction resources, and an action plan when they are ready.<br>• If trafficking exit is wanted, support with multidisciplinary care and service coordination for complex needs.<br>• Tailor interventions to strengths and resources.<br>• Address upstream and downstream health determinants.<br>• Facilitate access to mental health services. | • Create a catalogue of resources to meet the complex needs of survivors<br>• Establish a network of HCPs trained in anti-trafficking responses.<br>• Develop protocols and care pathways that reflect the constraints and resources of each setting:<br> • ER and urgent care settings should be able to address acute needs and refer to community care settings.<br> • Community settings should coordinate access to care for short, medium, and long-term exit needs.<br>• Provide longitudinal access to programs and services. |
| **Prevention** | • Participate in outreach efforts toward at-risk youth and sex workers.<br>• Collaborate with community stakeholders and survivors to raise awareness. | • Raise public consciousness of sex trafficking through awareness campaigns |

part of HCPs [22,23]. Our results show that HCPs are often unaware that they are encountering sex-trafficked individuals, and therefore are unprepared to conduct a proper assessment or to offer required resources. Surprisingly, this finding extended to sexual health clinics and community-based organizations, despite regular access by youth in sex work and/or the unhoused (i.e., groups most at risk). Participants remarked that HCPs who may have suspected abuse or trafficking did not act on their suspicions. This may stem from a lack of skills in screening and identifying sex-trafficked individuals. A study in the UK reported similarly that a lack of confidence in intervention strategies and making appropriate referrals for sex-trafficked individuals led to withheld assistance [24]. Stigmatization is also clear in survivors' accounts and was enacted through denial of care, provision of substandard care, and for some, verbal abuse, and humiliation. This was attributed to a lack of awareness which fosters misconceptions, bias, and prejudice [25]. Experiences of stigma and judgment resulted in poor health outcomes, non-disclosure, under-identification, and further alienation from HCPs. Distrust of HCPs from previous healthcare interactions becomes an additional barrier to receiving care for marginalized populations [26]. Together, these findings demonstrate the need to increase all HCPs' awareness and knowledge of trafficking cues to reduce under-identification and to implement training on screening and interventions for sex-trafficked individuals. Because our participants' demographic data differed from trafficking profiles presented in the literature

(e.g., all Canadian-born vs foreign-born), we caution that when developing protocols and training material, applying cues to potentially sex-trafficked individuals should reflect the general and local demographic profile.

A further novel finding was the more positive perceptions of care in community-based settings than in acute care settings, potentially explained by differences in workplace cultures. While emergency and urgent care are appropriate to address acute health conditions, the long wait times, HCP's workload, and poor continuity of care are challenges when serving people in situations of vulnerability [27,28]. Inadequate staffing often adds to these challenges and results in shallower interactions with patients. Similarly, high patient turnover in walk-in private clinics seemed to pre-empt the establishment of a trusting relationship. On the other hand, sexual health clinics or community-based organizations tend to offer more one-on-one care, case management including referral and scheduling appointments, long-term relationships with patients, and established community partnerships designed for vulnerable populations [28]. This finding is consistent with other studies examining services for populations in situations of vulnerability, wherein strong evidence pointed to better patient-reported health outcomes with regular engagement in community-based primary care settings [28–30]. From this result, sex-trafficked individuals would benefit from case management and coordination of care in community-based primary care settings.

Since perception of care was found to be reliant on behavioral competence and approaches of HCP, the positive reviews associated with sexual health clinics and community-based organizations may also be attributed to their greater exposure to marginalized groups and/or the use of sexual health and harm reduction approaches used in these settings. Both approaches promote a non-judgmental and non-coercive provision of high-quality care centered on the individual's needs and lifestyle choices and reducing harmful consequences of certain behaviors [31,32]. These approaches to care combined with the HCP's position as a trusted authority figure create a unique opportunity to build rapport and encourage disclosure. To achieve these goals, a trusting relationship was said to be achievable by providing a safe and private environment for the sex-trafficked individuals and initiating a respectful, therapeutic exchange.

Once trust is achieved, HCPs ought to ask questions to query trafficking based on observed cues. Contrary to findings with similar groups (e.g., women who experienced intimate partner violence), study participants recommended avoiding direct questioning on trafficking. Sex-trafficked individuals might not identify as such, so questions should aim to stimulate awareness of trafficking and facilitate disclosure (e.g., ask about work, partners, and personal life and point out signs of trafficking). Querying trafficking should be done routinely and as needed. HCP should also be properly supported since clinical events or trauma accounts can affect HCPs' well-being [33].

Once identified as trafficked individuals, HCPs should be able to provide them with resources to address their complex health needs, regardless of the desire to exit sex work. For this to occur, HCPs and organizations must be aware of existing resources and establish care pathways and referral networks with a comprehensive list of experts and services to address upstream and downstream needs [34]. Consistent with harm reduction approaches, protocols and guidelines should recognize trafficked individuals' agency in wanting to exit completely or remain in sex work: services should be unconditional. HCPs should provide resources to address current needs, including non-trafficking-related needs. Once they are ready to exit, interventions should be tailored to the individuals' strengths and resources and include multidisciplinary care and service coordination. Finally, access to mental health services should be facilitated by outreach initiatives and a readily available network of mental health providers trained in interventions with sex-trafficked individuals and survivors. Services should be

longitudinal as the exiting process can be long and mental health consequences lasting (see the complete list of recommendations in Table 1).

## Study limitations

This study focused on a very hard-to-reach population, only reachable through the collaboration with the Les Survivantes program of the SPVM. Our sample did not include males or international trafficking survivors as we were unable to access a more diverse pool of survivors. Trafficking survivors who did not use the services of the Les Survivantes program and exited on their own were not included as well. Furthermore, many survivors were actively involved in anti-trafficking advocacy through the program which may represent a participation bias. Therefore, our findings may not be generalizable to all sex trafficking survivors. Recruitment and data collection was stopped on pragmatic grounds rather than at the point of data saturation point, which explains the small sample size of this study. Despite this, although nuances within sub-themes were still emerging towards the end of data analysis, the themes themselves were being replicated indicating a level of completeness.

The regional focus may also limit the generalization of findings to healthcare settings elsewhere in Canada or other countries. All participants accessed healthcare services in an urban center and their experiences may not reflect the experiences of those who accessed care in rural settings. Finally, this study provided a broad range of time after exiting trafficking which may represent recall bias.

## Conclusion

Despite these limitations, this study provides new and important insights into how to support trafficked individuals during interactions with the healthcare system. We found that perceptions of care were context-dependent and largely based on HCP's approaches and behavioral competence which are modifiable through training and raising awareness. Our findings highlight the importance of centering care around trafficked individuals' agency and needs, and we propose that interventions require multidisciplinary community collaboration for better coordination and continuation of care. Broadly translated, our findings indicate that HCPs are largely unaware of trafficking and there is a need for more evidence-based research to inform protocol and care delivery for this population in an extreme situation of vulnerability. Future research should be geared towards the creation and validation of screening tools to identify trafficked individuals as well as best practice guidelines on how to intervene to help them in the best ways possible.

## Supporting information

**S1 Appendix.**
(PDF)

## Acknowledgments

We acknowledge the contribution of Josée Mensales and Romy Verge-Boudreau from the Survivors program of the "Service de Police de la Ville de Montréal (SPVM)". We also acknowledge the support of the Ministry of Higher Education and the Quebec Order of Nurses.

## Author Contributions

**Conceptualization:** Johane Lorvinsky, John Pringle, Françoise Filion, Anita J. Gagnon.

**Data curation:** Johane Lorvinsky.

**Formal analysis:** Johane Lorvinsky.

**Investigation:** Johane Lorvinsky.

**Methodology:** Johane Lorvinsky, John Pringle, Françoise Filion, Anita J. Gagnon.

**Project administration:** Johane Lorvinsky.

**Resources:** Johane Lorvinsky, John Pringle.

**Supervision:** John Pringle, Françoise Filion, Anita J. Gagnon.

**Validation:** Françoise Filion, Anita J. Gagnon.

**Writing – original draft:** Johane Lorvinsky.

**Writing – review & editing:** John Pringle, Françoise Filion, Anita J. Gagnon.

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
