## [Decision Letter · Decision Letter 0]

14 Jul 2022

PONE-D-22-13333Sex trafficking survivors’ experiences with the healthcare system during exploitation: a qualitative studyPLOS ONE

Dear Dr. Gagnon,

Thank you for submitting your manuscript to PLOS ONE. After careful consideration, we feel that it has merit but does not fully meet PLOS ONE’s publication criteria as it currently stands. Therefore, we invite you to submit a revised version of the manuscript that addresses the points raised during the review process.

 In your revised submission, please respond to each comment from each reviewer, particularly amending and clarifying the results and discussion sections. We look forward to receiving your revised submission.

We look forward to receiving your revised manuscript.

Kind regards,

Amy Michelle DeBaets, PhD

Academic Editor

PLOS ONE

Journal Requirements:

Reviewers' comments:

Reviewer's Responses to Questions

**Comments to the Author**

1. Is the manuscript technically sound, and do the data support the conclusions?

Reviewer #1: Partly

Reviewer #2: Yes

2. Has the statistical analysis been performed appropriately and rigorously? 

Reviewer #1: Yes

Reviewer #2: N/A

3. Have the authors made all data underlying the findings in their manuscript fully available?

Reviewer #1: Yes

Reviewer #2: Yes

4. Is the manuscript presented in an intelligible fashion and written in standard English?

Reviewer #1: Yes

Reviewer #2: Yes

5. Review Comments to the Author

Reviewer #1: General comments:

Please consider reducing victim centered language—use survivor when appropriate, and if the term survivor cannot be used consider changing phrases like “victims of sex trafficking” to “individuals who have experienced sex trafficking” or “trafficked individuals” etc.

Abstract

Abstract background, lines 23/24: Please be more specific regarding the purpose of your study here. Healthcare experiences of survivors is a broad topic, and this has also been explored extensively in the existing literature. Consider revising this sentence to include exactly what aspects of healthcare experiences this study explored. This will help prospective readers understand what exactly the study was looking at and also understand why this study is novel/different from the existing literature.

Abstract methods lines 27/28: This sentence is a bit vague, if you would like to describe further what you did in the study I would suggest being more specific regarding what “clinical phenomena” was studied. This can also be taken out and you may simply state that you used “interpretive description” to interpret your results.

Abstract results: This section should report objective data that was found. It currently reads more like a discussion. I would suggest summarizing the objective findings in a similar way to how the results section of your manuscript was written

Abstract conclusion, lines 39/40: Please be more specific here. What challenges experienced by trafficked individuals in accessing care/resources were found? What new insights and advice did the study provide? There has been prior literature on all of these topics—the reader should understand what exactly your study found and how it is novel/different from what exists in the literature.

Abstract conclusion, line 41: would suggest removing “Fortunately” from this sentence.

Introduction

Introduction, line 44: Please use a more current source for your estimates of human trafficking. I would suggest using the International Labour Organization (ILO) estimates which were most recently published in 2017: https://www.ilo.org/global/publications/books/WCMS_575479/lang--en/index.htm

Introduction, lines 56 – 78 (Review of the Literature section): I would suggest expanding your literature review to include the prior existing literature on healthcare experiences of trafficked individuals. This will contextualize your study, and will help us understand what the current state of literature is on healthcare experiences and why the present study is different/what it adds to the literature. Some suggestions for sources to include:

https://onlinelibrary.wiley.com/doi/abs/10.1111/cch.12759

https://pubmed-ncbi-nlm-nih-gov.laneproxy.stanford.edu/32190715/

https://pubmed.ncbi.nlm.nih.gov/34419901/

https://www.ncbi.nlm.nih.gov/pmc/articles/PMC5072917/

Introduction, line 66: Please fix typo (two commas)

Introduction, lines 95-97: Consistent with my comments on the abstract, I would suggest making your study aim statement more specific to identify what experiences with healthcare the study aimed to explore.

Methods

Methods, line 105: “…people sex trafficked…” -- Consistent with comment above, would suggest standardizing language when referring to trafficked individuals.

Results

Results, line 173: I would suggest including the demographics table in the body of the manuscript itself as “Table 1” rather than as an appendix—this is vital information for practitioners hoping to utilize the data from your study. Also would suggest reporting the full range of current ages of the participants, and defining “visible minority,” “CEGEP,” and exit “LOT” as a foot note of the table.

Results, Figure 1: I would suggest expanding this figure to include more bullet points regarding the main/novel findings for each theme. The reader should be able to get an idea of the findings of the study from looking at your main tables and figures.

Results, line 191: Consider re-naming the sub-theme “Restricted access to care.” Within this sub-theme, you demonstrate a variation in access amongst survivors surveyed, not just restrictions. It would make more sense to re-name this sub-theme to encompass this variation, not just the restrictions.

Results, lines 199 – 203: I would also suggest re-structuring the paragraph to better summarize findings from individuals who were and were not restricted better and expand on the survivor experiences that were not restricted rather than a quick mention at the end of this paragraph.

Results, lines 206 – 207: Please introduce this quote (ex: As participant 5 stated, “If you’re run down…”)

Results, line 207: Please clarify what is meant by “go through the healthcare system” – does this mean seek healthcare?

Novel findings to highlight: trust in healthcare providers, unaccompanied by traffickers

Discussion, Line 401: If recommendations were reported by survivors, I would suggest including this in the results section instead. If recommendations were synthesized by the authors, please add more information in the discussion on how these were created.

Discussion, Lines 427 – 435: This section includes information on various health care settings (community vs acute care settings)—the raw data regarding this does not seem to be emphasized in the results section. I would recommend including reporting on this in the results section if this is going to be touched on in the discussion.

Discussion general comments: I would recommend better emphasis on what makes this study truly novel, since some of these themes have already been reported in the literature.

Limitations: Please also address the small sample size of the study as a limitation.

Conclusion, Lines 494 – 496: This recommendation is very vague and general, would recommend being more specific on “tools” and “screening and interventions” and “goals of helping in the best ways possible,” or consider removing this sentence.

Reviewer #2: Hearing from survivors themselves was exciting and their statements are congruent with other research, healthcare issues, and trafficking situations encountered by anti-trafficking non-profit organizations. I found a few run-in sentences with the use of "and" in excess. I would revise those to enhance fluidity and comprehension of the material.

6. PLOS authors have the option to publish the peer review history of their article (what does this mean?). If published, this will include your full peer review and any attached files.

Reviewer #1: No

Reviewer #2: **Yes: **Dr. Marielle Combs

---

## [Author Response · Author response to Decision Letter 0]

13 Dec 2022

Response to reviewer’s comments

Thank you for the opportunity to address the reviewer’s comments and revise the manuscript. Please find point-by-point answers to comment below. Revisions in the manuscript are marked using the “track changes” feature in Word.

Author response (AR): Revised as requested

AR: Revised as requested. Please see the explanation in our cover letter.

Reviewer #1: General comments:

Please consider reducing victim centered language—use survivor when appropriate, and if the term survivor cannot be used consider changing phrases like “victims of sex trafficking” to “individuals who have experienced sex trafficking” or “trafficked individuals” etc.

AR: Following your recommendation, victim-centered language was removed. The term “victim” was replaced by “sex trafficked individuals” and “trafficked individuals”.

Abstract

Abstract background, lines 23/24: Please be more specific regarding the purpose of your study here. Healthcare experiences of survivors is a broad topic, and this has also been explored extensively in the existing literature. Consider revising this sentence to include exactly what aspects of healthcare experiences this study explored. This will help prospective readers understand what exactly the study was looking at and also understand why this study is novel/different from the existing literature.

AR: The purpose of the study was revised the following way:

In this study, we examined the experiences of accessing care of sex trafficking survivors in the Greater Montreal area and their interactions with HCPs to inform trafficking education programs for HCPs and allied health professionals regarding the needs of this patient population. 

Abstract methods lines 27/28: This sentence is a bit vague, if you would like to describe further what you did in the study I would suggest being more specific regarding what “clinical phenomena” was studied. This can also be taken out and you may simply state that you used “interpretive description” to interpret your results.

AR: The sentence was replaced by:

We used interpretive description to understand the lived experiences of trafficked individuals with direct applications to clinical education and care.

Abstract results: This section should report objective data that was found. It currently reads more like a discussion. I would suggest summarizing the objective findings in a similar way to how the results section of your manuscript was written

AR: This section was amended like so:

Our results revealed that trafficked individuals accessing care present with a fragile trust in HCPs and how HCPs have many opportunities to conduct investigative examinations and query trafficking. Trafficked individuals’ initial trust in HCPs can be strengthened by non-judgemental approaches or damaged by stigmatizing conduct, serving to isolate further and alienate this patient population. Health professionals’ attitudes combined with healthcare settings’ cultures of care (i.e., community vs emergency) and exposure to marginalized groups were key influencers of survivors’ perception of healthcare interactions. The findings also emphasized the importance of routinely querying trafficking through sensitive psychosocial questioning based on observation of trafficking cues. Survivors reported a list of trafficking cues to recognize and emphasize the importance of trust as a condition of disclosure. Finally, survivors identified the need for exit planning to be centered around trafficked individuals’ agency and holistic needs and for streamlined community-based multidisciplinary collaboration to serve this patient population better. 

Abstract conclusion, lines 39/40: Please be more specific here. What challenges experienced by trafficked individuals in accessing care/resources were found? What new insights and advice did the study provide? There has been prior literature on all of these topics—the reader should understand what exactly your study found and how it is novel/different from what exists in the literature.

AR: Revised as requested. The conclusion was modified:

Our results highlight that most challenges experienced by trafficking survivors in accessing care and resources are modifiable through HCP education and training. Our study also provides new insights and concrete advice to improve care and support throughout the exiting process. We argue that health care services for this population be modeled harm reduction approaches that focus on victims’ agency and needs, independent of their desire to exit trafficking. We emphasize the urgent need for proper case management and intersectoral and multidisciplinary care coordination in community-based settings as well as facilitated access to mental health support

Abstract conclusion, line 41: would suggest removing “Fortunately” from this sentence.

AR: Revised as requested.

Introduction

Introduction, line 44: Please use a more current source for your estimates of human trafficking. I would suggest using the International Labour Organization (ILO) estimates which were most recently published in 2017: https://www.ilo.org/global/publications/books/WCMS_575479/lang--en/index.htm

AR: The previous text was replaced by the following:

“According to the International Labour Organization’s (ILO) most recent and rather conservative estimates, 89 million people were victims of human trafficking worldwide between 2012 and 2017, with 40.3 million actively being trafficked in 2016”.

Introduction, lines 56 – 78 (Review of the Literature section): I would suggest expanding your literature review to include the prior existing literature on healthcare experiences of trafficked individuals. This will contextualize your study, and will help us understand what the current state of literature is on healthcare experiences and why the present study is different/what it adds to the literature. Some suggestions for sources to include:

https://onlinelibrary.wiley.com/doi/abs/10.1111/cch.12759

https://pubmed-ncbi-nlm-nih-gov.laneproxy.stanford.edu/32190715/

https://pubmed.ncbi.nlm.nih.gov/34419901/

https://www.ncbi.nlm.nih.gov/pmc/articles/PMC5072917/

AR: The literature review was updated:

Individuals in sex trafficking experience a wide array of physical, sexual, and psychological violence (6-8). They commonly report incidences of assault, threats, intimidation, humiliation, and degradation, as well as rape and restriction of freedom (9). Sequelae from these harms are wide-ranging and include several physical, mental, and reproductive health problems (6, 9-11). Most commonly, individuals under sexual exploitation experience acute injuries, untreated health conditions, anxiety and depression, post-traumatic stress symptoms, sexually transmitted and blood-borne infections (STBBIS), numerous pregnancies and terminations (spontaneous or induced) and other sexual health issues (6, 9-11). The significant medical and mental health needs may result in their seeking medical treatment(9, 12). Research in other developed countries has shown that trafficked individuals have limited access to health care as they are typically monitored, isolated, and denied access to their identifying documents such as their health insurance cards (5, 6). Yet, most victims receive medical treatment at some point during trafficking (9). These points of contact with healthcare are often missed opportunities for HCPs to identify victims and provide services and resources to address their needs (9, 13). When they meet healthcare providers (HCP), trafficked individuals seldom disclose their abuse due to fear of their trafficker, low self-esteem and feelings of shame, and lack of knowledge about judicial and healthcare systems (5, 6). Concurrently, HCPs are ill prepared to identify victims of trafficking (9). Under-identification means less access to services and resources to escape the trafficking situation (5). The lack of evidence-based guidelines or protocols on how to engage with and care for this population, as well as the limited resources to meet the needs of survivors, further alienate trafficked individuals and may result in impaired future health outcomes for survivors (10, 14). Training for HCP needs to include clear guidance to identify people in trafficking and steps to intervene with information on relevant resources and specialized trafficking services (15).

Introduction, line 66: Please fix typo (two commas)

AR: The second comma was removed

Introduction, lines 95-97: Consistent with my comments on the abstract, I would suggest making your study aim statement more specific to identify what experiences with healthcare the study aimed to explore.

AR: The aim of the study was clarified like so:

In conjunction with the directors of “Les Survivantes” program, we conducted this needs-based study in Montreal, Quebec to explore the experiences of sex trafficking survivors with the health care system. Hearing from survivors exiting sex trafficking and positioning them justifiably as experts provides insight into their lived experiences to inform future education programs for health care professionals tailored to the needs of this underserved population. Of particular interest are survivors’ ability to perceive, seek and engage in health care, the missed cues such as signs and symptoms that may have been conveyed to healthcare workers indicating that their patient is being sex trafficked and their advice for HCP intervention once identified. Missed cues represent missed opportunities to meet the person where she is at in her journey, the difficulty to provide appropriate care and a barrier to promote the patient’s liberation. 

Few studies have sought to explore trafficking survivors’ experiences with the healthcare system in Canada during their exploitation(16), and to the author’s knowledge, none were specific to Greater Montreal area. Since experiences of sex trafficking are often specific to population demographics and the larger societal context/geographic region, this study will equip providers to recognize trafficking cues specific to the Greater Montreal, an area cited as a hotspot for sexual exploitation in North America (17).

Methods

Methods, line 105: “…people sex trafficked…” -- Consistent with comment above, would suggest standardizing language when referring to trafficked individuals.

AR: Replaced by “…Sex trafficked individuals…”

Results

Results, line 173: I would suggest including the demographics table in the body of the manuscript itself as “Table 1” rather than as an appendix—this is vital information for practitioners hoping to utilize the data from your study. Also would suggest reporting the full range of current ages of the participants, and defining “visible minority,” “CEGEP,” and exit “LOT” as a foot note of the table.

AR: Demographics table was moved and became table 1. A legend key was added to the footnote:

Table 1 - a. Visible minority: non-white in skin color; b. CEGEP = Collège d'enseignement général et professionnel (similar to a technical college or vocational school); c. LOT = length of time.

Results, Figure 1: I would suggest expanding this figure to include more bullet points regarding the main/novel findings for each theme. The reader should be able to get an idea of the findings of the study from looking at your main tables and figures.

AR: Thank you for this suggestion. Although we have not modified this Figure, we have added tables (2, 3 and appendix E) to support our findings.

Results, line 191: Consider re-naming the sub-theme “Restricted access to care.” Within this sub-theme, you demonstrate a variation in access amongst survivors surveyed, not just restrictions. It would make more sense to re-name this sub-theme to encompass this variation, not just the restrictions.

AR: Revised as requested. The subtheme was renamed “Variation in access to care”.

Results, lines 199 – 203: I would also suggest re-structuring the paragraph to better summarize findings from individuals who were and were not restricted better and expand on the survivor experiences that were not restricted rather than a quick mention at the end of this paragraph.

AR: The paragraph was modified like this:

Survivors interviewed reported varying degrees of freedom of care, which was largely determined by the limitations imposed by traffickers. Most related how traffickers exerted power and control over their decisions, including their ability to seek care, through a variety of nefarious means such as violence, threats, continuing monitoring, financial control, and gaslighting. These tactics led to a loss of autonomy for participants, who were unable to make their own decisions, including seeking any form of help beyond the limitations imposed. 

If you are under the influence of someone, you can't make your own decisions, your life is governed by another person, and if you are being abused, the person will always ask you where you are, what you are doing, and who you are with. (P2)

In one case, the survivor was completely banned to seek medical treatment and their interactions with healthcare was accidental: “Well, I went straight to the hospital because I was in Quebec City…. and it was a car accident because otherwise I would never have been able to go to the hospital” (P1). Others were allowed to seek treatment for emergencies such as when a health issue impeded trafficking activities or in times of severe crisis, or to maintain their sexual health. As participant 5 stated, “If you’re run down, he has no choice but to take you to the doctor. Otherwise, you don't make money,". Some survivors were able to have regular access to a community-based organization offering safe drug paraphernalia and sexual health services for STBBI prevention and treatment : “they expect you to have good sexual health because they won't stop you from getting checked” (P2). These contact with healthcare were still under close monitoring for some survivors, through constant text messaging or fear of repercussions if they took too long to return. For one survivor, the trafficker went as far as accompanying her to the hospital. Survivors who did not relay explicit limitations to healthcare utilization also mainly sought treatment for sexual and reproductive health emergencies. Participants agreed that most people in sexual exploitation will seek healthcare during their exploitation given the risks inherent to sex work.

Results, lines 206 – 207: Please introduce this quote (ex: As participant 5 stated, “If you’re run down…”)

AR: Revised as requested

Results, line 207: Please clarify what is meant by “go through the healthcare system” – does this mean seek healthcare?

Novel findings to highlight: trust in healthcare providers, unaccompanied by traffickers

AR: The sentence was replaced by:

“Participants agreed that most sex trafficked individuals will seek healthcare during their exploitation…”

Discussion, Line 401: If recommendations were reported by survivors, I would suggest including this in the results section instead. If recommendations were synthesized by the authors, please add more information in the discussion on how these were created.

AR: The sentence was revised to reflect that recommendations are derived from survivors accounts :

Recommendations were derived from survivors’ advice and are summarized in Table 2. 

A table was also added in order to complement survivors advices already in the results section that inspired recommendations.

Discussion, Lines 427 – 435: This section includes information on various health care settings (community vs acute care settings)—the raw data regarding this does not seem to be emphasized in the results section. I would recommend including reporting on this in the results section if this is going to be touched on in the discussion.

AR: Data concerning this section can be found in the subtheme context-dependent perception.

Discussion general comments: I would recommend better emphasis on what makes this study truly novel, since some of these themes have already been reported in the literature.

AR: Please notice that in the discussion, when presenting a novel finding, we tried to highlight it with words such as “Surprisingly” or “A further novel finding”

Limitations: Please also address the small sample size of the study as a limitation.

AR: This limitation was addressed the following way:

Recruitment and data collection was stopped on pragmatic grounds rather than at the point of data saturation point, which explains the small sample size of this study. Despite this, although nuances within sub-themes were still emerging towards the end of data analysis, the themes themselves were being replicated indicating a level of completeness.

Conclusion, Lines 494 – 496: This recommendation is very vague and general, would recommend being more specific on “tools” and “screening and interventions” and “goals of helping in the best ways possible,” or consider removing this sentence.

AR: This sentence was modified:

Future research should be geared towards creation and validation screening tools to identify trafficked individuals as well as best practice guidelines on how to intervene with the goal of helping them in the best ways possible.

Reviewer #2: Hearing from survivors themselves was exciting and their statements are congruent with other research, healthcare issues, and trafficking situations encountered by anti-trafficking non-profit organizations. I found a few run-in sentences with the use of "and" in excess. I would revise those to enhance fluidity and comprehension of the material.

AR: Sentences were shortened and rearranged, when possible, to enhance fluidity.

---

## [Decision Letter · Decision Letter 1]

14 Jun 2023

PONE-D-22-13333R1Sex trafficking survivors’ experiences with the healthcare system during exploitation: a qualitative studyPLOS ONE

Dear Dr. Gagnon,

Thank you for submitting your manuscript to PLOS ONE. After careful consideration, we feel that it has merit but does not fully meet PLOS ONE’s publication criteria as it currently stands. Therefore, we invite you to submit a revised version of the manuscript that addresses the points raised during the review process.

 Please see below for recommendations of minor edits that will ensure that the terminology in the manuscript closely reflects the current state of the science.

We look forward to receiving your revised manuscript.

Kind regards,

Michelle L. Munro-Kramer, PhD, CNM, FNP-BC

Academic Editor

PLOS ONE

Journal Requirements:

Additional Editor Comments:

Thank you for your resubmission and careful attention to the previous reviewer comments. As noted by Reviewer #2, the paper attended to all previous reviewer comments. The syntax and language used to describe human trafficking is very important and proper use is essential to prevent further myths and misunderstanding among healthcare providers and academics. There are a few minor changes to the revised paper noted below that should be considered prior to publication in order to ensure that the terminology accurately reflects the state of the science:

1) Abstract: Second sentence of paragraph three uses the term "investigative examination". I am not sure exactly what the authors mean here (forensic, comprehensive)? But I believe the term investigative denotes that healthcare providers are investigating situations surrounding human trafficking or violence, which is not part of a healthcare providers' role. I would recommend changing this term to more accurately reflect the role of healthcare providers.

2) Introduction - first paragraph: Human trafficking is an umbrella term encompassing labor trafficking, sex trafficking, organ trafficking, forced marriage, etc. The statistics from ILO include all of these different forms of trafficking, yet they are presented as only representing sex trafficking. Please provide a sentence noting the broad definition of human trafficking and stating that this paper will focus exclusively on sex trafficking.

3) Page 6, line 96-98 - The abbreviation HCP is used before it is introduced and then is not used consistently throughout the paper.

4) Page 6, line 102-103 - The term "escape" is used here. As the remainder of this paper focuses on survivor self-agency on determining what is best, consideration should be given to avoid the narrative of "rescuing" human trafficking victims by using terms like "escape" and instead focused on providing them the resources and support they desire.

5) Page 7, line 127 - The term "she" is used to describe a sex trafficking survivor. I would consider avoiding gendered language or provide justification for it's use.

6) Page 7, line 130 - Change "author's knowledge" to "authors' knowledge"

7) Page 7, line 133 - There is a great deal of data to refute current rankings or hotspots of human trafficking because of the poor prevalence data available internationally (due to the reasons you note in the introduction). Please use caution with the term "hotspot" and consider providing justification based on data.

8) Page 11, Ethical considerations - Were participants offered resources or support if the interview process brought up emotions or concerns that they wanted to discuss with someone?

9) Table 2 - please provide a key to define RAMQ

10) Page 20, line 348 - The abbreviation LCHC is never used again. I would remove it.

11) Page 28, line 501 - Should "exiting services" be "existing services"?

12) Page 344, line 619 - Change "All participant accessed" to "All participants accessed"

Reviewers' comments:

Reviewer's Responses to Questions

**Comments to the Author**

1. If the authors have adequately addressed your comments raised in a previous round of review and you feel that this manuscript is now acceptable for publication, you may indicate that here to bypass the “Comments to the Author” section, enter your conflict of interest statement in the “Confidential to Editor” section, and submit your "Accept" recommendation.

Reviewer #2: All comments have been addressed

2. Is the manuscript technically sound, and do the data support the conclusions?

Reviewer #2: Yes

3. Has the statistical analysis been performed appropriately and rigorously? 

Reviewer #2: Yes

4. Have the authors made all data underlying the findings in their manuscript fully available?

Reviewer #2: Yes

5. Is the manuscript presented in an intelligible fashion and written in standard English?

Reviewer #2: Yes

6. Review Comments to the Author

Reviewer #2: The authors addressed all areas of concern presented by the reviewers. The manuscript will benefit those working against human trafficking and persons being trafficked.

7. PLOS authors have the option to publish the peer review history of their article (what does this mean?). If published, this will include your full peer review and any attached files.

Reviewer #2: **Yes: **Dr. Marielle Combs

---

## [Author Response · Author response to Decision Letter 1]

29 Jul 2023

Thank you for the giving us the opportunity to submit a revised draft of our manuscript. We are grateful for the time and effort that you dedicated to give us insights and feedback. We have incorporated most of the suggestions you made. Please find point-by-point answers to comments below. Revisions in the manuscript are marked using the “track changes” feature in Word.

1) Abstract: Second sentence of paragraph three uses the term "investigative examination". I am not sure exactly what the authors mean here (forensic, comprehensive)? But I believe the term investigative denotes that healthcare providers are investigating situations surrounding human trafficking or violence, which is not part of a healthcare providers' role. I would recommend changing this term to more accurately reflect the role of healthcare providers.

As suggested, the word investigative was replaced by comprehensive.

2) Introduction - first paragraph: Human trafficking is an umbrella term encompassing labor trafficking, sex trafficking, organ trafficking, forced marriage, etc. The statistics from ILO include all of these different forms of trafficking, yet they are presented as only representing sex trafficking. Please provide a sentence noting the broad definition of human trafficking and stating that this paper will focus exclusively on sex trafficking.

Thank you for this suggestion. In the first paragraph, we mention that the ILO statistics represent human trafficking, not sex trafficking: “According to the International Labour Organization’s (ILO) most recent and rather conservative estimates, 89 million people were victims of human trafficking worldwide between 2012 and 2017, with 40.3 million actively being trafficked in 2016.”. However, in Canada, the number of police-reported cases are mostly sex trafficking, which we define later: “In Canada, the consensus is that human trafficking in and through the country is much more prevalent than suggested by previous estimates or the 1700 police-reported cases from 2009–2018, of which commercial sexual exploitation or sex trafficking is the largest subset.” As the other forms of trafficking are not as present in Canada according to reports and since the paper focuses on sex trafficking, we chose not to give the broader definition of human trafficking which has been widely discussed in the literature.

3) Page 6, line 96-98 - The abbreviation HCP is used before it is introduced and then is not used consistently throughout the paper.

Thank you for pointing this out. This was revised. 

4) Page 6, line 102-103 - The term "escape" is used here. As the remainder of this paper focuses on survivor self-agency on determining what is best, consideration should be given to avoid the narrative of "rescuing" human trafficking victims by using terms like "escape" and instead focused on providing them the resources and support they desire.

We think this is an excellent suggestion. The word escape was replaced with exit.

5) Page 7, line 127 - The term "she" is used to describe a sex trafficking survivor. I would consider avoiding gendered language or provide justification for it's use.

As suggested, we replaced “she” with “they”.

6) Page 7, line 130 - Change "author's knowledge" to "authors' knowledge"

Thank you for pointing out this typo. We have corrected it.

7) Page 7, line 133 - There is a great deal of data to refute current rankings or hotspots of human trafficking because of the poor prevalence data available internationally (due to the reasons you note in the introduction). Please use caution with the term "hotspot" and consider providing justification based on data.

We agree that the term should be removed. This was changed to “a destination for sexual tourism in North America”.

8) Page 11, Ethical considerations - Were participants offered resources or support if the interview process brought up emotions or concerns that they wanted to discuss with someone?

Yes, we added the following to highlight this fact: “Care was taken to prevent and mitigate distress during the interview (see Appendix D in S1) and participants had access to mental health care through the survivors’ program. They were strongly encouraged to utilize them if they needed additional support post interview.”

9) Table 2 - please provide a key to define RAMQ

This was removed entirely and replaced by “Quebec health insurance card”.

10) Page 20, line 348 - The abbreviation LCHC is never used again. I would remove it.

As suggested, this was removed.

11) Page 28, line 501 - Should "exiting services" be "existing services"?

This was meant to be “exiting services” and so it was left as is.

12) Page 344, line 619 - Change "All participant accessed" to "All participants accessed"

Thank you for pointing out this typo. We have corrected it.

---

## [Editor Report · Decision Letter 2]

2 Aug 2023

Sex trafficking survivors’ experiences with the healthcare system during exploitation: a qualitative study

PONE-D-22-13333R2

Dear Dr. Gagnon,

We’re pleased to inform you that your manuscript has been judged scientifically suitable for publication and will be formally accepted for publication once it meets all outstanding technical requirements.

Kind regards,

Michelle L. Munro-Kramer, PhD, CNM, FNP-BC, FAAN

Academic Editor

PLOS ONE

Additional Editor Comments (optional):

Thank you for addressing the additional edits. I look forward to seeing this manuscript published.
---

## [Editor Report · Acceptance letter]

21 Aug 2023

PONE-D-22-13333R2 

Sex trafficking survivors’ experiences with the healthcare system during exploitation: a qualitative study 

Dear Dr. Gagnon:

I'm pleased to inform you that your manuscript has been deemed suitable for publication in PLOS ONE. Congratulations! Your manuscript is now with our production department. 

Kind regards, 

on behalf of

Dr. Michelle L. Munro-Kramer 

Academic Editor

PLOS ONE